# Synthesis and Biological Assessment of 4,1-Benzothiazepines with Neuroprotective Activity on the Ca^2+^ Overload for the Treatment of Neurodegenerative Diseases and Stroke

**DOI:** 10.3390/molecules26154473

**Published:** 2021-07-24

**Authors:** Lucía Viejo, Marcos Rubio-Alarcón, Raquel L. Arribas, Manuel Moreno-Castro, Raquel Pérez-Marín, María Braun-Cornejo, Martín Estrada-Valencia, Cristóbal de los Ríos

**Affiliations:** 1Instituto-Fundación Teófilo Hernando and Departamento de Farmacología y Terapéutica, Universidad Autónoma de Madrid, C/Arzobispo Morcillo, 4, 28029 Madrid, Spain; lucia.viejo@estudiante.uam.es (L.V.); marcru02@ucm.es (M.R.-A.); raquel.arribas@urjc.es (R.L.A.); manmoren@ucm.es (M.M.-C.); raquelpm4@gmail.com (R.P.-M.); m.brauncornejo@gmail.com (M.B.-C.); 2Instituto de Investigación Sanitaria, Servicio de Farmacología Clínica, Hospital Universitario de La Princesa, 28006 Madrid, Spain

**Keywords:** CGP37157, benzothiazepines, neuroprotection, Ca^2+^ overload, oxidative stress, mitochondria

## Abstract

In excitable cells, mitochondria play a key role in the regulation of the cytosolic Ca^2+^ levels. A dysregulation of the mitochondrial Ca^2+^ buffering machinery derives in serious pathologies, where neurodegenerative diseases highlight. Since the mitochondrial Na^+^/Ca^2+^ exchanger (NCLX) is the principal efflux pathway of Ca^2+^ to the cytosol, drugs capable of blocking NCLX have been proposed to act as neuroprotectants in neuronal damage scenarios exacerbated by Ca^2+^ overload. In our search of optimized NCLX blockers with augmented drug-likeness, we herein describe the synthesis and pharmacological characterization of new benzothiazepines analogues to the first-in-class NCLX blocker CGP37157 and its further derivative ITH12575, synthesized by our research group. As a result, we found two new compounds with an increased neuroprotective activity, neuronal Ca^2+^ regulatory activity and improved drug-likeness and pharmacokinetic properties, such as clog *p* or brain permeability, measured by PAMPA experiments.

## 1. Introduction

Calcium ions act as very versatile intracellular second messengers. Calcium originates a plethora of cell responses, such as neurotransmitters exocytosis, muscle contraction or gene transcription activation [1]. The cytosolic Ca^2+^ concentration ([Ca^2+^]_c_) is an essential parameter that needs to be finely regulated. Calcium homeostasis depends on the quantity of Ca^2+^ uptake through plasmalemmal transporters and its release from intracellular organelles [1]. The Ca^2+^ entry to the cytosol mainly occurs via ligand or voltage-gated ionic channels. The first ones comprise the *N*-methyl-d-aspartate-sensitive glutamate receptors (NMDAR) [2], the α-amino-3-hydroxy-5-methyl-4-isoxazolpropionic acid-sensitive glutamate receptors (AMPAR) [3], and some subtypes of the acetylcholine nicotinic receptors [4]. Among the second ones, voltage-gated Ca^2+^ channels (VGCC) outstand [5]. Ca^2+^ ions can be released to cytosol from endoplasmic reticulum (ER) through inositol-1,4,5-triphosphate receptors (IP3R) or ryanodine receptors [6], and from mitochondria through both mitochondrial Na^+^/Ca^2+^ (NCLX) and H^+^/Ca^2+^ exchangers [7]. Cells are capable of decreasing [Ca^2+^]_c_ with Ca^2+^-chelating proteins, such as Calbindin-D28K and parvalbumin, or with plasmalemmal transporters like the Na^+^/Ca^2+^ exchanger [8]. Furthermore, ER and mitochondria uptake Ca^2+^ through the sarcoplasmic/endoplasmic reticulum Ca^2+^ ATPase [1] and the mitochondrial Ca^2+^ uniporter [9]. The malfunction of any of these mechanisms gives rise to cell Ca^2+^ overload or deficit, in sensitive areas of excitable cells, which promote situations of cell morbidity [10].

The impairment of the [Ca^2+^]_c_ balance has been widely documented in several central and vascular diseases. In particular, this is a common pathological event in neurodegenerative diseases, where Ca^2+^ dyshomeostasis is more exacerbated than the minor Ca^2+^ deregulation observed in a normal aging. In Alzheimer’s disease (AD), the Ca^2+^ dyshomeostasis hypothesis points out that Ca^2+^ dysregulation in intracellular stores, plasma membrane transporters, and buffering mechanisms accelerate the progression of the disease. Therefore, the Ca^2+^ dysregulation evidenced in AD causes functional impairments, leading to brain damage and cognitive deficit [11]. Moreover, Ca^2+^ imbalance has been strongly correlated with several hallmarks of AD, as amyloid β-peptide (Aβ) oligomers can form pores in the membranes permeable to Ca^2+^ [12]. In addition, soluble Aβ oligomers induce Ca^2+^ dyshomeostasis through NMDAR overstimulation, driving to synapsis loss in mice brains [13]. Interestingly, the first approved drug for AD within the last 18 years (approved in 7 June 2021 by FDA), aducanumab [14], reduced Ca^2+^ overload in animal models [15]. Besides, the other non-anticholinesterasic drug approved for AD is other Ca^2+^ regulator, the NMDAR blocker memantine [16]. Various Ca^2+^ permeable channels such as NMDAR [17], AMPAR [18], VGCC [19] or ionotropic purinergic receptors 7 [20,21] are also implicated in the progression of the amyotrophic lateral sclerosis. These Ca^2+^ channels participate in the amplification of the neuronal lesion, for instance, triggered by the glutamate released by activated microglia, and where the mitochondrial dysfunction seems to be the main target [22,23]. The α-synuclein, chief component of the Lewy bodies of dopaminergic neurons compromised in Parkinson’s disease, forms small aggregates (protofibrils) able to generate ionic pores in the lipid membrane [24], and to induce an uncontrolled Ca^2+^ entry to neurons [25]. Additionally, the continuous firing exerted by dopaminergic neurons needs high fluxes of Ca^2+^, what makes these neurons particularly vulnerable to Ca^2+^ dyshomeostasis [26]. The polyglutamined huntingtin protein, characteristic of the Huntington’s disease, binds directly to IP3R, and induces a massive Ca^2+^ release from the endoplasmic reticulum [27]. The pathological huntingtin protein also affects both NMDA and VGCC permeability to Ca^2+^ [28,29]. As far as stroke, it is considered as an “acute” neurodegeneration process, where cerebral ischemic injury shares physiopathological mechanisms of neuronal death with neurodegenerative diseases. Many of these mechanisms control Ca^2+^ dysregulation as a key event generated by the large exposure to glutamate [30] and the subsequent recruitment of NMDAR and AMPAR. Thus, compounds studied for ischemic brain injury, such as melatonin, have featured their neuroprotective actions through the expression of Ca^2+^-binding proteins [31].

In summary, all these observations reinforce the clinical study of Ca^2+^ channels antagonists and the search of new drugs capable of regulating [Ca^2+^]_c_. These remain a therapeutic challenge, where innovative approaches focused on interacting with intracellular organelles buffering Ca^2+^ are getting more attention every day. Under this premise, our research group studies new 4,1-benzothiazepine derivatives analogues to **CGP37157** (Figure 1) [32] with important neuroprotective actions in several in vitro models of neurodegeneration and stroke. Their pharmacological activity involves biological targets handling [Ca^2+^]_c_, as VGCC [33], NCLX [34], or the Ca^2+^ homeostasis modulator 1 [35]. Hence, these compounds can be considered as a privileged family of Ca^2+^ modulators. Our main goal aimed to improve their pharmacokinetic properties, increasing their aqueous solubility while keeping their neuroprotective profile. Then, through some isosteric replacements, compounds with better Lipinsky’s rules of five parameters were obtained without compromising pharmacodynamics (Appendix A) [36].

Figure 1 shows lead compounds of previously described families of 4,1-benzothiazepines. Based on these results, we herein described a newly-synthesized **CGP37157** analogues presenting chemical descriptors that afforded the best pharmacological activities in previous studies, as well as the replacement by pyridine of the pending phenyl ring, in order to supply less lipophilic drugs. Alternatively, isomers of **ITH12575** were also prepared with the goal of assessing the effect of shifting the position of the isopropyl group on the pharmacological activity. In addition, the use of dimethylamine as an isoster of chlorine has been probed, which would also enhance pharmacokinetic profile of derivatives.

## 2. Results and Discussion 

### 2.1. Chemistry

The preparation of 4,1-benzothiazepine derivatives was achieved through two main reaction steps, namely o-lithiation (Scheme 1, step a) and S-alkylation (Scheme 1, step b), following experimental procedures previously reported with slight modifications [34]. 

Non-commercial isopropylbenzaldehydes, necessary to prepare intermediates **5** and **6**, were synthesized from the corresponding bromoisopropylbenzene by metal–halogen exchange with ^n^BuLi in presence of DMF as a source of carbonyl (Appendix A). Boc-protected p-chloroaniline [35] suffered NH–deprotonation by ^t^BuLi dropwise addition at −78 °C. To promote the subsequent ortho-metallation process, temperature was elevated up to −20 °C. Then, the resulting organolithium intermediates underwent nucleophilic addition on the selected aromatic aldehydes to produce diarylcarbinol derivatives **5**–**7** in medium to low yields. The addition of aryllithium species over arylaldehydes was affected by various obstacles, being the main one product degradation by an intramolecular nucleophilic attack of the generated alcohol over the carbamate, similarly to what was previously reported [38]. With the aim of increasing polarity and aqueous solubility by the incorporation of aromatic nitrogen atoms, some pyridylaldehydes were used as substrates for the addition reaction, which produced intermediates **8**–**11**. Next, preparation of compounds **12**–**18** was executed by treatment with an excess of methyl thioglycolate in refluxing of trifluoroacetic acid (TFA; 85 °C). In this acidic environment, several reactions proceed in one pot. A SN1 reaction occurs due to removal of H_2_O and attack of the thiol reagent over the stabilized carbocation, along with the release of the Boc-protected amine. Thus, the newly free amine attacks the pending methyl ester in an intramolecular fashion, to yield the corresponding benzothiazepines **12**–**18** (Scheme 1) in moderate to excellent yields. In previous works, substrates possessing aromatic nitrogen atoms needed the use of alternative reagents and additional reaction steps [36]. By contrast, the 5-(pyridin-N-yl)benzothiazepines **15**–**18** were obtained in good yields and under similar conditions to those for the synthesis of the 5-arylbenzothiazepines **12**–**14**. As shown in Appendix A, pyridine-bearing derivatives **15**–**18** present a better aqueous solubility, according to their calculated partition coefficient logarithm (clog *p*) value that are between 2.5 and 3.2.

A second strategy to increase polarity of 4,1-benzothiazepines was explored. It consisted of replacing the chlorine atom at position 7 by a dimethylamino group. The Boc-protected N,N-dimethyl-p-phenylenediamine **19** (Scheme 2) (Appendix A) was subjected to the same conditions of ortho-lithiation, and the nucleophilic addition over o-isopropylbenzaldehyde led to the formation of intermediate **20** in moderated yield (33%). In this case, the ortho-lithiation reaction followed by the nucleophilic addition gave lower yields with several other tested aromatic aldehydes (e.g., with o-chlorobenzaldehyde, yielding the corresponding intermediate in a 12% yield), presumably due to an enhanced intramolecular addition of carbinol to the carbamate, with removal of tert-butanol. Then, the reaction of **20** with methyl thioglycolate in TFA at reflux afforded the corresponding 7-(dimethylamino)-5-(2-(propan-2-yl)phenyl-1,5-dihydro-4,1-benzothiazepin-2(3*H*)-one **21** with a 38% yield (Scheme 2).

Attempting to increase yields, we replaced the Boc- group with pivaloyl (2,2-dimethylpropanoyl) as a protecting group, which had offered a reliable alternative for the preparation of pyridothiazepines [36]. Thus, N,N-dimethyl-p-phenylenediamine was protected at the free amine with pivaloyl chloride using triethylamine as a base in excellent yields. The subsequent ortho-lithiation reaction of **22** was promoted with ^t^BuLi at −20 °C as commented above, and its addition to selected aromatic aldehydes furnished the corresponding open intermediates **23**–**25** in low to good yields, but with a more straightforward purification than that experimented with the Boc-protected intermediate **20** and analogues. Formation of the benzothiazepine scaffold needed higher temperatures, i.e., 110 °C, to avoid the isolation of S-alkylated non-cycled intermediate. Thus, we obtained in one-pot products **26**–**28** in low yields (Scheme 3). Nevertheless, this entails a significant optimization since the removal of pivaloyl required in previous reports a three-step protocol [36] (removal of pivaloyl protecting group, S-alkylation, and finally, a LiHMDS-catalyzed cyclization).

The replacement of chlorine by dimethylamine at C7 afforded a slight reduction of the predicted clog *p* (Appendix A). Some other derivatives proposed to be tested were benzothiazepines unsubstituted at C7. These new analogues were expected to be more lipophilic than the head **CGP37157**, but their clog *p* also lowered, about one unit. Consequently, we synthesized new derivatives starting from aniline, which was firstly protected with pivaloyl chloride, and the subsequent addition of aromatic aldehydes via ortho-lithiation provided the expected intermediates **30** and **31**. With the goal of probing alternative methodologies, preparation of fluorinated intermediate **30** was carried out through ortho-lithiation with ^n^BuLi in smooth conditions. Although yield was not disappointing in this example (43%), the reaction with other aldehydes were unsuccessful. Moreover, the disturbing presence of the by-product derived from the addition of n-butyllithium to p-fluorobenzaldehyde in the crude product moved us back to use ^t^BuLi as metallating agent. We achieved addition to the bulky o-anisaldehyde by inducing ortho-metallation of **29** with ^t^BuLi, affording intermediate **31** in low yield. The treatment with methyl thioglycolate at 100 °C underwent deprotection, S-alkylation, and cyclization in one pot; however, this treatment produced very low yields (Scheme 4).

The alternative three-independent-steps methodology was carried out as described [36]. Nevertheless, it did not entail an improvement in terms of overall chemical yield compared with the one-pot preparation, in addition to increasing the time required and expenses.

Finally, we appreciated that 4-aminopyridine could become an eligible starting point for the synthesis of new pyridothiazepines with enhanced polarity, different to the previously described pyrido [2,3-*e*][1,4] thiazepines [36], as they would give rise to interesting pyrido[4,3-*e*][1,4] thiazepine bicycles. Protection of 4-aminopyridine with pivaloyl chloride was conducted in good yield at lower temperature (−20 °C), to avoid the formation of diacylated product, observed at rt. Regular ortho-lithiation conditions and the subsequent addition to benzaldehyde, conducted this time at −20 °C unlike previous reactions, provided the corresponding intermediate **35** in excellent yield (90%). Then, the one-pot reaction using methyl thioglycolate in TFA at high temperature (up to 115 °C, including the use of sealed tube) failed. Alternatively, we recruited the three steps plan. Hence, pivaloyl group was removed under basic aqueous conditions (KOH in ethanol/H_2_O reflux) and the free amino alcohol **36** suffered nucleophilic substitution of methyl thioglycolate aided by TFA under CH_2_Cl_2_ reflux, obtaining **36** in low yield, presumably due to the scarce contribution of the pyridine ring to the stability of the intermediate carbocation. Lastly, cyclization to form the desired pyrido[4,3-*e*][1,4] thiazepine **38** was accomplished by deprotonation of the amine with LiHMDS in THF (Scheme 5). The global yield of these three chemical steps reached a decent 22% (13% from aniline, five reaction steps). Intriguingly, the incorporation of an aromatic nitrogen at position 7 gave rise to a pyridothiazepine derivative with a very good clog *p* value (1.93, Appendix A), much better than those of the previously reported pyrido[2,3-*e*][1,4] thiazepines [36].

### 2.2. Pharmacology

#### 2.2.1. Assessment of per se Toxicity of the New Derivatives in Cell Cultures

Firstly, we aimed to ensure that the new synthesized compounds had no negative effect over the viability of our cellular models. Thus, SH-SY5Y neuroblastoma cells and rat cortical neurons were exposed to compounds at increasing concentrations (1, 3, 10, and 30 μM) (Figure 2). First, experiments of viability in SH-SY5Y cells in presence of tested compounds evidenced that compound **13**, bearing an isopropyl group at meta position, showed a remarkable cytotoxic profile at 30 μM (Figure 2), reason why **13** was discontinued from the pharmacological analysis. Despite compound **12** did not produce a dramatic loss of cell viability, this new molecule was not further evaluated on the grounds that some solubility problems were appreciated. Indeed, both compounds are highly lipophilic benzothiazepines analogues to **ITH12575**, as confirmed by their clog *p* (Appendix A). The other compounds herein described did not affect cell viability in a statistically significant fashion. Regarding cortical neurons, compounds **18**, **26**, **27**, **28**, and **33** slightly reduced cell viability in a significant manner at 30 μM. Thus, we paid attention to their therapeutic range in experiments using cortical neurons.

Overall, these experiments helped us to stablish proper testing concentrations. For instance, considering that the neuroprotective experiments need long incubation period, the selected concentration was 10 μM. In the case of the modulation of neuronal Ca^2+^ oscillations, assays to evaluate IC_50_ in the micromolar range seemed reliable, as they are acute assays.

#### 2.2.2. Effect of Compounds on the Membrane Depolarization-Induced and NMDA-Induced Ca^2+^ Influx

Subsequently, we assessed the ability of the compounds to minimize altered Ca^2+^ elevations into cells in our two cellular models. On one hand, we subjected SH-SY5Y neuroblastoma cells to a high extracellular K^+^ concentration (70 mM), which evokes cell depolarization and VGCC opening [33]. This experimental model mimics what occurs in a plethora of neurodegenerative diseases, where Ca^2+^ overload is a common event and mitochondria play a valuable role to buffer such Ca^2+^ elevations. Should our compounds slow the rate of Ca^2+^ release from mitochondria, they would improve the capacity of neurons to handle the neurotoxic cytosolic Ca^2+^ accumulation. Thus, most of the synthesized compounds, preincubated to SH-SY5Y cells at the concentration of 10 μM, showed an amenable mitigating effect of the depolarization-stimulated [Ca^2+^]_c_ elevation in SH-SY5Y cells by about 50% (Figure 3). Compounds **18**, **21**, and **32** decreased Ca^2+^ signaling at 10 μM in a similar manner or even better than the reference compound nifedipine (10 μM) (Figure 3). This appreciable Ca^2+^ antagonist feature allowed us to calculate their IC_50_, defined as the concentration of compound capable of halving the 70 mM K^+^-evoked Ca^2+^ elevation. For that purpose, SH-SY5Y cells were incubated with the test compounds at several concentrations from 1 to 30 μM and then stimulated with high K^+^ to induce depolarization. In this situation, increasing concentrations of compounds produced a proportional decline of Ca^2+^ uptake, allowing us to draw concentration–response curves (Figure 3). Those compounds prevailing in the single concentration experiments, **18**, **21**, and **32**, presented the lowest IC_50_ of the family, 11.5, 5.4, and 9.2 μM, respectively.

On the other hand, NMDAR is one the principal pathways of Ca^2+^ overload reported in neurodegenerative diseases [39]. Indeed, a unique non-anticholinesterasic drug prescribed for AD is the NMDAR antagonist memantine [16]. For this reason, we considered of note to assess whether these potential NCLX blockers could also mitigate the altered Ca^2+^ fluctuations derived from NMDAR stimulation. To achieve this aim, we chose the primary culture of the rat cortical embryonic neurons, as SH-SY5Y neuroblastoma cells have not operative NMDAR [40]. In this protocol, NMDA was administered at the concentration of 10 μM to rat cortical embryonic neurons to induce Ca^2+^ entry through the NMDAR. The subsequent [Ca^2+^]_c_ elevation was slightly reduced by compounds exposed to neurons at 10 μM (Figure 4), mostly by about 25%, except **21** and **27**, which showed a noticeable mitigation of the NMDA-elicited Ca^2+^ increase, even higher than that exerted by the reference compound memantine at 10 μM. Therefore, only compounds **21** and **27** offered IC_50_ values below 30 μM, being 6.2 and 18.5 μM, respectively.

#### 2.2.3. Effect of Compounds against Toxic Stimuli Related to Neurodegeneration

The neuroprotective effects of these compounds in in vitro models of neurodegeneration were evaluated against a cocktail of rotenone and oligomycin A (R/O, 30 μM/10 μM) and the glutamate exposure (50 μM). For the former, SH-SY5Y neuroblastoma cells were subjected to oxidative stress by administration of the stressor cocktail R/O, which disrupts the correct function of the mitochondrial electron transport chain by inhibiting complex I and V, respectively. It is worthwhile mentioning that this oxidative stress model related to aging produces, in terms of the mitochondrial impairment, neuronal Ca^2+^ dishomeostasis [41]. Hence, a potential NCLX partial blocker should efficiently counteract the neurotoxicity elicited by R/O. Thus, the effect of the compounds administered at 10 μM against the R/O damage was evaluated by the MTT reduction method. As it can be appreciated in Figure 5, dimethylamino analogues **21**, **26**, **27**, and **28** present a clear trend to protect against oxidative stress, as these compounds are able to reduce the cell death induced by R/O. Beside this family, pyridothiazepine **38** also showed a neuroprotective profile against oxidative stress.

Regarding the other neurodegeneration model, the effect against the excitotoxicity produced by high glutamate concentration was selected, considering that during the progression of neurodegenerative diseases, neurons lose their ability to control glutamate concentrations, while this neurotransmitter keeps in the synaptic cleft, producing cell death by over-activating NDMAR [42]. Similar to the fluorescence experiments of NMDA-evoked [Ca^2+^]_c_ oscillations, the cell culture used for these assays were rat cortical embryonic neurons, considering their susceptibility to glutamate administration. In these experiments, compounds elicited a different pattern of neuroprotection. For instance, compounds **21**, **32**, and **33** presented a statistically significant dissipation of the glutamate-elicited neurotoxicity. Again, compound **38** nicely recovered cell viability of the glutamate-challenged cortical neurons.

#### 2.2.4. Blood–Brain Barrier Crossing Ability

Considering the potential use of these compounds as drugs targeting the central nervous system (CNS), we evaluated their ability to penetrate the blood–brain barrier (BBB) using the parallel artificial membrane permeation assay (PAMPA) prediction method. The in vitro permeabilities (Pe) of compounds to CNS were obtained by executing these experiments in the presence of the lipid extract of a porcine brain as a model of human BBB, as described in the experimental section. Negative and positive reference drugs were used to assess the correct development of each assay. Thus, most of the benzothiazepine derivatives described in this work were capable of crossing the BBB by passive diffusion, showing Pe values higher than 4, a widely accepted value that defines the limit for passive diffusion [43]. As it can be appreciated in Figure 6, only compounds **14**, **26**, and **32** did not reach this Pe value. In the case of **14**, the presence of the highly charged nitro group presumably hinders its passive diffusion.

## 3. Materials and Methods

### 3.1. General

Commercially available reagents for synthesis or biological experiments were purchased from Sigma-Aldrich/Merck (Madrid, Spain). Solvents, with analytical grade, were obtained from VWR/Avantor (Barcelona, Spain) and freshly distilled when used for reactions. Thin layer chromatography (TLC) under silica gel was used to monitor the evolution of reactions by UV light exposure at 254 nm. Reactions were carried out under Schlenk conditions (vacuum/argon purges and argon atmosphere). Further purifications of compounds were conducted by automatized flash chromatography ISOLERA One (Biotage, Uppsala, Sweden), using pre-charged “SNAP” columns. Melting points, without correction, were obtained with a SMP-10 apparatus (Stuart, Stone, Staffordshire, UK). MS spectra were acquired in a QSTAR ABSciex apparatus (Applied Biosystems, Foster City, CA, USA). We acquired 1H and 13C NMR spectra in a AVANCE 300 MHz station (Bruker, Billerica, MA, USA), shown as parts per million (ppm) calibrated by the residual proton signal from the deuterated solvents. The purity of the final compounds was evaluated by elemental analysis in a CHNS-932 station (Leco, St Joseph, MI, USA), where mass percentages were within the 0.4% compared with the calculated value, or by HRMS in an ABSciex QSTAR spectrometer under the high-resolution configuration with electrospray as the ionization source, where *m*/*z* values were within the 5 ppm compared with the calculated value.

### 3.2. General Procedure for the Synthesis of Intermediates **5**–**11**, **20**, **26**–**28**, **31**, and **35**

General procedure was similar to what has been described, ref. [34] with slight modifications. To a solution of the Boc- or Piv-protected anilines **4** [35], **19**, **22**, and **29** (Appendix A) (1 equiv) in dry THF (4–8 mL/mmol) at −78 °C under Ar atmosphere, ^t^BuLi (1.7 M in hexanes, 2.7 equiv) was added dropwise. When the addition was complete, the mixture was stirred for 15 min at −78 °C, then it was warmed up to −20 °C and stirred for 2 h more. After this time, the reaction mixture was cooled down to −78 °C and a solution of the corresponding arylaldehyde (1.1 equiv) in THF (5 mL) was slowly added, stirring the resulting mixture at −78 °C until no longer evolution was observed (2–5 h) by TLC. Reaction was interrupted by slow addition of H_2_O (15 mL) and allowed to reach rt. Afterwards, it was extracted with diethyl ether (3 × 30 mL) and combined organic layer was washed with brine (3 × 50 mL), dried over anhydrous MgSO_4_, filtered, and evaporated, obtaining an oil that was purified by automated flash chromatography using ethyl acetate/n-hexane mixtures as eluent.

#### 3.2.1. 4-Chloro-2-[hydroxy(4′-isopropylphenyl)methyl]-N-tert-butoxycarbonylaniline (**5**)

Following the General procedure 4.2, reaction of 4-chloro-N-tert-butoxycarbonylaniline **4** [35] (2.28 mmol, 519 mg) with p-isopropylbenzaldehyde (Appendix A) (2.51 mmol, 372 mg) [^t^BuLi (6.16 mmol)] yielded **5** (341 mg, 40%). ^1^H NMR (300 MHz, CDCl_3_) δ 7.81 (d, J = 8.70 Hz, 1H, Ar), 7.66 (s, 1H, NH), 7.30 (m, 6H, Ar), 7.08 (d, J = 2.5 Hz, 1H, H3), 5.84 (s, 1H, CHOH), 3.60 (bs, 1H, OH), 2.94 [m, J = 6.9 Hz, 1H, CH(CH_3_)_2_], 1.47 [s, 9H, C(CH_3_)_3_], 1.28 [d, J = 6.9 Hz, 6H, CH(CH_3_)_2_].

#### 3.2.2. 4-Chloro-2-[hydroxy(3′-isopropylphenyl)methyl]-N-tert-butoxycarbonylaniline (**6**)

Following the General procedure 4.2, reaction of 4-chloro-N-tert-butoxycarbonylaniline **4** [35] (0.66 mmol, 150 mg) with m-isopropylbenzaldehyde (Appendix A) (0.72 mmol, 107 mg) [^t^BuLi (1.78 mmol)] yielded **6** (72 mg, 29%). ^1^H NMR (300 MHz, CDCl_3_) δ 7.84 (d, J = 8.9 Hz, 1H, Ar), 7.55 (s, 1H, NH), 7.35–7.15 (m, 4H, Ar), 7.10–7.05 (m, 2H, Ar), 5.85 (s, 1H, CHOH), 3.50 (bs, 1H, OH), 2.92 [m, J = 6.8 Hz, 1H, CH(CH_3_)_2_], 1.45 [s, 9H, C(CH_3_)_3_], 1.25 [d, J = 6.8 Hz, 6H, CH(CH_3_)_2_].

#### 3.2.3. 4-Chloro-2-[hydroxy(4′-nitrophenyl)methyl]-N-tert-butoxycarbonylaniline (**7**)

Following the General procedure 4.2, reaction of 4-chloro-N-tert-butoxycarbonylaniline **4** [35] (3.01 mmol, 685 mg) with p-nitrobenzaldehyde (3.31 mmol, 500 mg) [^t^BuLi (8.13 mmol)] yielded **7** (264 mg, 23%). ^1^H NMR (300 MHz, CDCl_3_) δ 8.20 (d, J = 9.0 Hz, 2H, Ar), 7.69 (d, J = 8.6 Hz, 1H, Ar), 7.55 (d, J = 9.0 Hz, 2H, Ar), 7.48 (s, 1H. NH), 7.31 (m, 1H, Ar), 7.09 (d, J = 2.5 Hz, 1H, H3), 5.93 (s, 1H, CHOH), 4.80 (bs, 1H, OH), 1.43 [s, 9H, C(CH_3_)_3_].

#### 3.2.4. 4-Chloro-2-[hydroxy(pyridin-4-yl)methyl]-N-tert-butoxycarbonylaniline (**8**)

Following the General procedure 4.2, reaction of 4-chloro-N-tert-butoxycarbonylaniline **4** [35] (4.24 mmol, 965 mg) with isonicotinaldehyde (4.67 mmol, 500 mg) [^t^BuLi (11.45 mmol)] yielded **8** (618 mg, 44%). ^1^H NMR (300 MHz, CDCl_3_) δ 8.29 (d, J = 6.2 Hz, 2H, Ar), 8.04 (s, 1H, NH), 7.77 (d, J = 8.8 Hz, 1H, Ar), 7.34 (d, J = 6.2 Hz, 2H, Ar), 7.28 (dd, J = 8.7, 2.4 Hz, 1H, Ar), 7.10 (d, J = 2.4 Hz, 1H, H3), 5.78 (s, 1H, CHOH), 4.70 (bs, 1H, OH), 1.41 [s, 9H, C(CH_3_)_3_].

#### 3.2.5. 4-Chloro-2-[hydroxy(pyridin-3-yl)methyl]-N-tert-butoxycarbonylaniline (**9**)

Following the General procedure 4.2, reaction of 4-chloro-N-tert-butoxycarbonylaniline **4** [35] (4.24 mmol, 965 mg) with nicotinaldehyde (4.67 mmol, 500 mg) [^t^BuLi (11.45 mmol)] yielded **9** (718 mg, 51%). ^1^H NMR (300 MHz, CDCl_3_) δ 8.42 (s, 1H, Ar), 8.30 (m, 1H, Ar), 8.18 (s, 1H, NH), 7.75 (m, 2H, Ar), 7.24 (m, 2H, Ar), 7.10 (d, J = 2.3 Hz, 1H, H3), 5.86 (s, 1H, CHOH), 4.90 (bs, 1H, OH), 1.43 [s, 9H, C(CH_3_)_3_].

#### 3.2.6. 4-Chloro-2-[hydroxy(pyridin-2-yl)methyl]-N-tert-butoxycarbonylaniline (**10**)

Following the General procedure 4.2, reaction of 4-chloro-N-tert-butoxycarbonylaniline **4** [35] (4.24 mmol, 965 mg) with picolinaldehyde (4.67 mmol, 500 mg) [^t^BuLi (11.45 mmol)] yielded **10** (394 mg, 28%). ^1^H NMR (300 MHz, CDCl_3_) δ 8.52 (d, J = 6.0 Hz, 1H, Ar), 8.38 (s, 1H, NH), 7.75 (d, J = 8.6 Hz, 1H, Ar), 7.67 (m, 1H, Ar), 7.24 (m, 4H, Ar), 5.79 (s, 1H, CHOH), 4.93 (bs, 1H, OH), 1.42 [s, 9H, C(CH_3_)_3_].

#### 3.2.7. 4-Chloro-2-[(3-chloropyridin-4-yl)hydroxymethyl]-N-tert-butoxycarbonylaniline (**11**)

Following the General procedure 4.2, reaction of 4-chloro-N-tert-butoxycarbonylaniline **4** [35] (1.93 mmol, 439 mg) with 3-isonicotinaldehyde (2.11 mmol, 300 mg) [^t^BuLi (5.21 mmol)] yielded **11** (144 mg, 20%). ^1^H NMR (300 MHz, CDCl_3_) δ 8.44 (d, J = 5.1 Hz, 1H, Ar), 8.38 (s, 1H, Ar), 7.67 (d, J = 5.1 Hz, 1H, Ar), 7.55 (s, 1H, NH), 7.53 (d, J = 8.7 Hz, 1H, Ar), 7.22 (dd, J = 8.7, 2.5 Hz, 1H, Ar), 6.89 (d, J = 2.5 Hz, 1H, H3), 6.03 (s, 1H, CHOH), 5.48 (bs, 1H, OH), 1.47 [s, 9H, C(CH_3_)_3_].

#### 3.2.8. 4-Dimethylamino-2-[hydroxy(2′-Isopropylphenyl)methyl]-N-tert-butoxycarbonylaniline (**20**)

Following the General procedure 4.2, reaction of 4-dimethylamino-N-tert-butoxycarbonylaniline **19** (Appendix A) (4.23 mmol, 1 g) with o-isopropylbenzaldehyde (4.65 mmol, 690 mg) [^t^BuLi (11.42 mmol)] yielded **20** (537 mg, 33%). ^1^H NMR (300 MHz, acetone-d_6_) δ 7.83 (bs, 1H, NH), 7.45 (m, 2H, Ar), 7.34–7.15 (m, 3H, Ar), 6.65 (dd, 1H, J = 8.7, 3.0 Hz, H5), 6.42 (s, 1H, CHOH), 6.22 (d, 1H, J = 3.0 Hz, H3), 4.94 (s, 1H, OH), 3.16 [m, J = 6.0 Hz, 1H, CH(CH_3_)_2_], 2.76 [s, 6H, N(CH_3_)_2_], 1.46 [s, 9H, C(CH_3_)_3_], 1.20 [d, J = 6.0 Hz, 3H, CH(CH_3_)], 0.98 [d, J = 6.0 Hz, 3H, CH(CH_3_)].

#### 3.2.9. 2-[(2′-Chlorophenyl)hydroxymethyl]-4-dimethylamino-N-tert-butylcarbonylaniline (**23**)

Following the General procedure 4.2, reaction of N-[4-(dimethylamino)phenyl]-2,2-dimethylpropanamide **22** (Appendix A) (2.26 mmol, 0.5 g) with o-chlorobenzaldehyde (0.28 mL, 318 mg, 2.26 mmol) [^t^BuLi (6.1 mmol)] yielded **23** (284 mg, 35%). ^1^H NMR (300 MHz, CDCl_3_) δ 7.96 (bs, 1H, NH), 7.57 (bd, J = 6.0, 1H, Ar), 7.4 (d, J = 9.0 Hz, 1H, H6), 7.3–7.20 (m, 3H, Ar), 6.65 (dd, J = 9.0, 3.0 Hz, H5), 6.38 (d, J = 3.0 Hz, 1H, H3), 6.05 (d, J = 2.7 Hz, 1H, CHOH), 4.11 (d, J = 2.7 Hz, 1H, OH), 2.81 [s, 6H, N(CH_3_)_2_], 1.25 [s, 9H, C(CH_3_)_3_].

#### 3.2.10. 4-Dimethylamino-2-[hydroxy(2′-Methoxyphenyl)methyl]-N-tert-butylcarbonylaniline (**24**)

Following the General procedure 4.2, reaction of N-[4-(dimethylamino)phenyl]-2,2-dimethylpropanamide **22** (Appendix A) (3.40 mmol, 750 mg) with o-methoxybenzaldehyde (0.45 mL, 509 mg, 3.74 mmol) [^t^BuLi (9.18 mmol)] yielded **24** (865 mg, 71%). ^1^H NMR (300 MHz, CDCl_3_) δ 8.79 (s, 1H, NH), 7.97 (d, 1H, J = 8.4 Hz, H6), 7.30 (m, 1H, Ar), 6.95 (d, J = 8.0 Hz, 1H, Ar), 6.87 (m, 2H, Ar), 6.74 (bd, J = 8.4 Hz, 1H, H5), 6.51 (d, J = 2.7 Hz, 1H, H3), 6.04 (d, J = 3.6 Hz, 1H, CHOH), 4.04 (d, 1H, J = 3.6 Hz, OH), 3.93 (s, 3H, OCH_3_), 2.86 [s, 6H, N(CH_3_)_2_], 1.06 [s, 9H, C(CH_3_)_3_].

#### 3.2.11. 4-Dimethylamino-2-[(4′-fluorophenyl)hydroxymethyl]-N-tert-butylcarbonylaniline (**25**)

Following the General procedure 4.2, reaction of N-[4-(dimethylamino)phenyl]-2,2-dimethylpropanamide **22** (Appendix A) (2.26 mmol, 0.5 g) with p-fluorobenzaldehyde (0.27 mL, 309 mg, 2.49 mmol) [^t^BuLi (6.10 mmol)] yielded **25** (390 mg, 50%). ^1^H NMR (300 MHz, CDCl_3_) δ 8.21 (s, 1H, NH), 7.82 (d, J = 9.0 Hz, 1H, H6), 7.27 (m, 2H, Ar), 6.98 (m, 2H, Ar), 6.72 (dd, J = 9.0, 3.0 Hz, 1H, H5), 6.50 (d, J = 3.0 Hz, 1H, H3), 5.75 (s, 1H, CHOH), 3.77 (s, 1H, OH), 2.88 [s, 6H, N(CH_3_)_2_], 1.04 [s, 9H, C(CH_3_)_3_].

#### 3.2.12. 2-[Hydroxy(2′-methoxyphenyl)methyl]-N-tert-butylcarbonylaniline (**31**)

Following the General procedure 4.2, reaction of N-phenyl-2,2-dimethylpropanamide **29** (Appendix A) (2.80 mmol, 0.5 g) with o-methoxybenzaldehyde (0.88 mL, 780 mg, 5.60 mmol) [^t^BuLi (7.00 mmol)] yielded **31** (215 mg, 25%). ^1^H NMR (300 MHz, CDCl_3_) δ 9.53 (s, 1H, NH), 8.48 (d, J = 8.4 Hz, 1H, H6), 7.67–7.06 (m, 7H, Ar), 6.31 (d, J = 4.3 Hz, 1H, CHOH), 4.46 (d, 1H, J = 4.3 Hz, OH), 4.17 (s, 3H, OCH_3_), 1.35 [s, 9H, C(CH_3_)_3_].

#### 3.2.13. N-[3-[hydroxy(phenyl)methyl]pyrydin-4-yl]-2,2-dimethylpropanamide (**35**)

Following the General procedure 4.2, reaction of 2,2-dimethyl-N-(4-pyridinyl)propanamide **34** (Appendix A) (4.21 mmol, 750 mg) benzaldehyde (0.47 mL, 491 mg, 4.62 mmol) [^t^BuLi (11.37 mmol)] yielded **35** (1.08 g, 90%). ^1^H NMR (300 MHz, CDCl_3_) δ 9.37 (s, 1H, NH), 8.37 (d, J = 5.7 Hz, 1H, Ar), 8.31 (d, J = 5.7 Hz, 1H, Ar), 7.99 (s, 1H, Ar), 7.36–7.30 (m, 5H, Ar), 5.85 (s, 1H, CHOH), 3.72 (s, 1H, OH), 1.07 [s, 9H, C(CH_3_)_3_].

### 3.3. Synthesis of 2-[(4′-Fluorophenyl)hydroxymethyl]-N-tert-butylcarbonylaniline (**30**)

Similar to what has been described for the use of ^t^BuLi [34], with modifications. To a solution of the Piv-protected aniline **29** (Appendix A) (2.82 mmol, 0.5 g) in dry THF (12 mL) at −78 °C under Ar atmosphere, ^n^BuLi (1.9 M in hexanes, 2.5 equiv, 7.05 mmol, 3.7 mL) was added dropwise. When the addition was complete, the mixture was stirred for 15 min at −78 °C, then it was warmed up to 0 °C and stirred for 4 h more. After this time, the reaction mixture was cooled down to −78 °C and a solution of p-fluorobenzaldehyde (0.62 mL, 0.7 g, 5.64 mmol) in THF (5 mL) was slowly added, and the reaction mixture was allowed to reach 0 °C for 2 h. Reaction was interrupted by slow addition of HCl 1N (12 mL). Afterwards, it was extracted with ethyl acetate (3 × 20 mL) and the combined organic layer was dried over anhydrous Na_2_SO_4_, filtered, and evaporated, obtaining an oil that was purified by automatized flash chromatography using ethyl acetate/n-hexane mixtures as eluent, yielding **30** as a yellow oil (360 mg, 43%). ^1^H NMR (300 MHz, CDCl_3_) δ 9.15 (s, 1H, NH), 8.37 (d, J = 8.1 Hz, 1H, Ar), 7.62–7.27 (m, 5H, Ar), 7.23 (m, 2H, Ar), 6.09 (s, 1H, CHOH), 4.22 (s, 1H, OH), 1.31 [s, 9H, C(CH_3_)_3_].

### 3.4. General Procedure for the Synthesis of 4,1-Benzothiazepines **12**–**18**, **21**, **26**–**28**

Similar to what has been described [34], with slight modifications. Methyl thioglycolate (6 equiv) and trifluoroacetic acid (TFA, 14 equiv) were added to the intermediates **5**–**11** or **20** under Ar. The mixture was stirred at 85–110 °C for 24 h. Then, it was dissolved in CH_2_Cl_2_ (30 mL) and washed subsequently with brine (30 mL), NaOH (1N, 30 mL), and brine (30 mL). The organic layer was dried over Na_2_SO_4_, filtered, and evaporated. The crude material was purified by automatized flash chromatography using ethyl acetate/n-hexane mixtures as eluent to give pure compounds.

#### 3.4.1. 7-Chloro-5-(4′-isopropylphenyl)-3,5-dihydro-4,1-benzothiazepin-2-(1*H*)-one (**12**)

Following the General procedure 4.4, reaction of **5** (0.66 mmol, 248 mg) [methyl thioglycolate (3.96 mmol, 420 mg), TFA (9.24 mmol, 1.05 g, 707 μL)] yielded **12** (140 mg, 64%) as a white solid. Mp: 240–243 °C. ^1^H NMR (300 MHz, CDCl_3_) δ 8.20 (s, 1H, NH), 7.39 (d, J = 8.1 Hz, 2H, Ar), 7.27 (m, 3H, Ar), 7.04 (d, J = 8.4 Hz, 1H, H9), 6.98 (d, J = 2.3 Hz, 1H, H6), 5.64 (s, 1H, H5), 3.34 (d, J = 12.2 Hz, 1H, H3a), 2.99 (d, J = 12.2 Hz, 1H, H3b), 2.94 [m, J = 6.9 Hz, 1H, CH(CH_3_)_2_], 1.28 [d, J = 6.9 Hz, 6H, CH(CH_3_)_2_]. ^13^C NMR (75.4 MHz, CDCl_3_) δ 170.6, 149.4, 136.9, 134.8, 133.4, 133.2, 129.4, 128.9, 128.7, 127.2, 125.4, 47.2, 34.0, 31.7, 24.1, 24.0. MS (API-ES+): *m*/*z* 332.09 [(M + H)^+^], 663.18 [(2M + Na)^+^]. Anal. C_18_H_18_ClNOS (C,H,N).

#### 3.4.2. 7-Chloro-5-(3′-isopropylphenyl)-3,5-dihydro-4,1-benzothiazepin-2-(1*H*)-one (**13**)

Following the General procedure 4.4, reaction of **6** (0.19 mmol, 71 mg) [methyl thioglycolate (1.15 mmol, 122 mg), TFA (2.66 mmol, 303 mg, 204 μL)] yielded **13** (21 mg, 33%) as a white solid. Mp: 218–220 °C. ^1^H NMR (300 MHz, CDCl_3_) δ 8.01 (s, 1H, NH), 7.30 (m, 2H, Ar), 7.25 (m, 3H, Ar), 6.98 (d, J = 8.4 Hz, 1H, H9), 6.90 (d, J = 2.3 Hz, 1H, H6), 5.59 (s, 1H, H5), 3.28 (d, J = 12.2 Hz, 1H, H3a), 2.92 (d, J = 12.2 Hz, 1H, H3b), 2.86 [m, J = 6.9 Hz, 1H, CH(CH_3_)_2_], 1.21 [d, J = 6.9 Hz, 3H, CH(CH_3_)], 1.19 [d, J = 6.9 Hz, 3H, CH(CH_3_)]. ^13^C NMR (75.4 MHz, CDCl_3_) δ 149.8, 136.8, 136.0, 134.8, 133.3, 129.0, 128.9, 128.7, 127.8, 127.0, 126.8, 125.4, 47.6, 34.2, 31.7, 24.2, 24.0. MS (API-ES+): *m*/*z* 332.10 [(M + H)^+^], 663.20 [(2M + Na)^+^]. HRMS: *m*/*z* (M + H)^+^ calcd. for C_18_H_18_ClNOS 332.0870, found, 332.0869.

#### 3.4.3. 7-Chloro-5-(4′-nitrophenyl)-3,5-dihydro-4,1-benzothiazepin-2-(1*H*)-one (**14**)

Following the General procedure 4.4, reaction of **7** (0.70 mmol, 265 mg) [methyl thioglycolate (4.20 mmol, 446 mg), TFA (9.80 mmol, 1.12 g, 750 μL)] yielded **14** (234 mg, >99%) as a white solid. Mp: 289–291 °C. ^1^H NMR (300 MHz, DMSOd_6_) δ 9.52 (s, 1H, NH), 8.25 (d, J = 8.8 Hz, 2H, Ar), 7.74 (d, J = 8.8 Hz, 2H, Ar), 7.45 (dd, J = 8.3, 2.4 Hz, 1H, Ar), 7.29 (d, J = 2.4 Hz, 1H, H6), 7.06 (d, J = 8.4 Hz, 1H, H9), 5.78 (s, 1H, H5), 3.14 (d, J = 12.2 Hz, 1H, H3a), 3.07 (d, J = 12.2 Hz, 1H, H3b). ^13^C NMR (75.4 MHz, DMSOd_6_) δ 168.4, 146.8, 146.4, 136.6, 135.6, 130.3, 129.8, 128.8, 128.7, 126.5, 123.5, 46.3, 30.5. MS (API-ES+): *m*/*z* 335.02 [(M + H)^+^]. Anal. C_15_H_11_ClN_2_O_3_S (C,H,N).

#### 3.4.4. 7-Chloro-5-(pyridin-4-yl)-3,5-dihydro-4,1-benzothiazepin-2-(1*H*)-one (**15**)

Following the General procedure 4.4, reaction of **8** (1.85 mmol, 619 mg) [methyl thioglycolate (11.10 mmol, 1.18 g), TFA (25.90 mmol, 2.95 g, 1.98 mL)] yielded **15** (369 mg, 69%) as a white solid. Mp: 224–226 °C. ^1^H NMR (300 MHz, DMSOd_6_) δ 9.51 (s, 1H, NH), 8.58 (m, 2H, Ar), 7.46 (m, 3H, Ar), 7.33 (d, J = 2.4 Hz, 1H, H6), 7.07 (d, J = 8.4 Hz, 1H, H9), 5.64 (s, 1H, H5), 3.15 (d, J = 12.2 Hz, 1H, H3a), 3.04 (d, J = 12.2 Hz, 1H, H3b). ^13^C NMR (75.4 MHz, DMSOd_6_) δ 168.4, 149.7, 147.8, 136.6, 135.3, 130.3, 128.9, 128.8, 126.5, 123.4, 45.9, 30.3. MS (API-ES+): *m*/*z* 291.03 [(M + H)^+^]. Anal. C_14_H_11_ClN_2_OS (C,H,N).

#### 3.4.5. 7-Chloro-5-(pyridin-3-yl)-3,5-dihydro-4,1-benzothiazepin-2-(1*H*)-one (**16**)

Following the General procedure 4.4, reaction of **9** (2.14 mmol, 716 mg) [methyl thioglycolate (12.84 mmol, 1.36 g), TFA (29.96 mmol, 3.42 g, 2.29 mL)] yielded **16** (476 mg, 76%) as a white solid. Mp: 224–226 °C. ^1^H NMR (300 MHz, DMSOd_6_) δ 9.62 (s, 1H, NH), 8.62 (d, J = 2.4 Hz, 1H, H2′), 8.55 (m, 1H, Ar), 7.93 (m, 1H, Ar), 7.43 (m, 2H, Ar), 7.13 (d, J = 2.4 Hz, 1H, H6), 7.07 (d, J = 8.4 Hz, 1H, H9), 5.70 (s, 1H, H5), 3.10 (AA’BB’, J = 12.5 Hz, 2H, H3). ^13^C NMR (75.4 MHz, DMSOd_6_) δ 168.5, 149.7, 149.0, 136.6, 136.5, 135.5, 133.8, 130.3, 128.7, 128.3, 126.3, 123.5, 44.2, 30.7. HRMS: *m*/*z* (M + H)^+^ calcd. for C_14_H_12_ClN_2_OS 291.0353, found, 291.0347.

#### 3.4.6. 7-Chloro-5-(pyridin-2-yl)-3,5-dihydro-4,1-benzothiazepin-2-(1*H*)-one (**17**)

Following the General procedure 4.4, reaction of **10** (1.18 mmol, 395 mg) [methyl thioglycolate (7.08 mmol, 751 mg), TFA (16.52 mmol, 1.88 g, 1.26 mL)] yielded **17** (200 mg, 58%) as a white solid. Mp: 215–217 °C. ^1^H NMR (300 MHz, CDCl_3_) δ 8.65 (m, 1H, Ar), 7.78 (dt, J = 7.7, 1.9 Hz, 1H, Ar), 7.65 (s, 1H, NH), 7.60 (d, J = 8.9 Hz, 1H, H6), 7.27 (m, 2H, Ar), 7.00 (m, 2H, Ar), 5.72 (s, 1H, H5), 3.26 (d, J = 12.5 Hz, 1H, H3a), 3.08 (d, J = 12.5 Hz, 1H, H3b). ^13^C NMR (75.4 MHz, CDCl_3_) δ 169.9, 156.6, 149.9, 137.3, 136.2, 134.8, 133.4, 129.0, 125.5, 123.9, 123.2, 49.3, 31.2. MS (API-ES+): *m*/*z* 291.03 [(M + H)^+^]. HRMS: *m*/*z* (M + H)^+^ calcd. for C_14_H_12_ClN_2_OS 291.0353, found, 291.0342.

#### 3.4.7. 7-Chloro-5-(3′-chloropyridin-4-yl)-3,5-dihydro-4,1-benzothiazepin-2-(1*H*)-one (**18**)

Following the General procedure 4.4, reaction of **11** (0.39 mmol, 144 mg) [methyl thioglycolate (2.34 mmol, 248 mg), TFA (5.46 mmol, 1.88 g, 1.26 mL)] yielded **18** (80 mg, 63%) as a white solid. Mp: 226–229 °C. ^1^H NMR (300 MHz, CDCl_3_) δ 8.64 (d, J = 6.0 Hz, 1H, H6′), 8.61 (s, 1H, H2′), 7.83 (d, J = 5.1 Hz, 1H, H5′), 7.51 (s, 1H, NH), 7.31 (dd, J = 2.4, 8.4 Hz, 1H, H8), 7.07 (d, J = 8.4 Hz, 1H, H9), 6.78 (d, J = 2.4 Hz, 1H, H6), 5.94 (s, 1H, H5), 3.28 (d, J = 12.3 Hz, 1H, H3a), 3.10 (d, J = 12.3 Hz, 1H, H3b). ^13^C NMR (75.4 MHz, CDCl_3_) δ 169.2, 150.0, 148.2, 148.1, 135.3, 135.0, 133.9, 133.7, 129.6, 128.5, 125.8, 125.4, 43.9, 31.2. HRMS: *m*/*z* (M + H)^+^ calcd. for C_14_H_11_Cl_2_N_2_OS 324.9964, found, 324.9953.

#### 3.4.8. 7-Dimethylamino-5-(2′-isopropylphenyl)-3,5-dihydro-4,1-benzothiazepin-2-(1*H*)-one (**21**)

Following the General procedure 4.4, reaction of **20** (0.26 mmol, 100 mg) [methyl thioglycolate (1.56 mmol, 165 mg), TFA (3.64 mmol, 415 mg, 279 μL)] yielded **21** (29 mg, 38%) as a white solid. Mp: 108–112 °C. ^1^H NMR (300 MHz, acetone-d_6_) δ 8.49 (s, 1H, NH), 7.78 (m, 1H, Ar), 7.40–7.25 (m, 3H), 7.00 (d, J = 8.4 Hz, 1H, H9), 6.66 (dd, J = 3.0, 8.4 Hz, 1H, H8), 6.09 (m, 2H, H6, H5), 3.33 (d, J = 12.0 Hz, 1H, H3a), 2.93 [m, J = 6.9 Hz, 1H, CH(CH_3_)_2_], 2.83 (d, J = 12.0 Hz, 1H, H3b), 2.72 [s, 6H, N(CH_3_)_2_], 1.19 [d, J = 6.9 Hz, 3H, CH(CH_3_)], 0.86 [d, J = 6.9 Hz, 3H, CH(CH_3_)]; ^13^C NMR (75.4 MHz, CDCl_3_) δ 169.8, 150.7, 148.0, 136.4, 135.4, 130.6, 129.3, 127.5, 127.0, 126.6, 125.8, 113.0, 112.4, 44.0, 40.6, 32.4, 24.5, 23.7. HRMS: *m*/*z* (M + H)^+^ calcd. for C_20_H_25_N_2_OS 341.1682, found, 341.1671.

#### 3.4.9. 5-(2′-Chlorophenyl)-7-dimethylamino-3,5-dihydro-4,1-benzothiazepin-2-(1*H*)-one (**26**)

Following the General procedure 4.4, reaction of **23** (0.39 mmol, 140 mg) [methyl thioglycolate (2.33 mmol, 247 mg), TFA (5.43 mmol, 619 mg, 415 μL)] yielded **26** (36 mg, 27%) as a white solid. Mp: 178–181 °C. ^1^H NMR (300 MHz, DMSOd_6_) δ 9.44 (s, 1H, NH), 7.81 (d, J = 7.5 Hz, 1H, Ar), 7.49 (m, 2H, Ar), 7.45 (m, 1H, Ar), 6.93 (d, J = 8.7 Hz, 1H, Ar), 6.66 (dd, J = 11.4, 5.4 Hz, 1H, H8), 5.95 (m, 2H, H6, H5), 3.16 (d, 1H, H3a), 2.90 (d, 1H, H3b), 2.75 [s, 6H, N(CH_3_)_2_]. ^13^C NMR (75.4 MHz, DMSOd_6_): δ 168.6, 148.9, 134.9, 132.9, 132.8, 131.0, 129.9, 129.8, 127.3, 126.3, 125.0, 112.0, 110.3, 43.5, 31.1. HRMS: *m*/*z* (M + H)^+^ calcd. for C_17_H_18_ClN_2_OS 333.0822, found, 333.0829. Anal. C_17_H_17_ClN_2_OS (C,H,N).

#### 3.4.10. 7-Dimethylamino-5-(2′-methoxyphenyl)-3,5-dihydro-4,1-benzothiazepin-2-(1*H*)-one (**27**)

Following the General procedure 4.4, reaction of **24** (0.70 mmol, 250 mg) [methyl thioglycolate (4.18 mmol, 444 mg), TFA (9.76 mmol, 1.11 g, 750 μL)] yielded **27** (24 mg, 10%) as a white solid. Mp: 233–236 °C. ^1^H NMR (300 MHz, CDCl_3_) δ 7.75 (d, J = 9.0 Hz, 1H, Ar), 7.33 (m, 1H, Ar), 7.11 (s, 1H, NH), 7.05 (m, 1H, Ar), 6,90 (m, 2H, Ar), 6.56 (dd, J = 9.0, 3.0 Hz, 1H, H8), 6.25 (d, J = 3.0 Hz, 1H, H6), 6.17 (s, 1H, H5), 3.70 (s, 3H, OCH_3_), 3.35 (d, J = 12.0 Hz, 1H, H3a), 2.95 (d, J = 12.0 Hz, 1H, H3b), 2.79 [s, 6H, N(CH_3_)_2_]. ^13^C NMR (75.4 MHz, DMSOd_6_): δ 170.6, 157.0, 150.0, 135.7, 130.8, 129.2, 125.7, 125.2, 125.2 120.4, 111.8, 111.1, 55.7, 40.6, 40.4, 31.7. Anal. C_18_H_21_N_2_O_2_S·H_2_O (C,H,N).

#### 3.4.11. 7-Dimethylamino-5-(4′-fluorophenyl)-3,5-dihydro-4,1-benzothiazepin-2-(1*H*)-one (**28**)

Following the General procedure 4.4, reaction of **25** (0.43 mmol, 150 mg) [methyl thioglycolate (2.61 mmol, 277 mg), TFA (6.09 mmol, 694 mg, 466 μL)] yielded **28** (25 mg, 18%) as a white solid. Mp: 219–222 °C. ^1^H NMR (300 MHz, DMSOd_6_) δ 9.14 (s, 1H, NH), 7.52 (m, 2H, Ar), 7.21 (m, 2H, Ar), 6.87 (d, J = 9.0 Hz, 1H, H9), 6.65 (dd, J = 9.0 Hz, J = 3.0 Hz, 1H, H8), 6.33 (d, J = 3.0, 1H, H6), 5.54 (s, 1H, H5), 3.05 (d, J = 12.0 Hz, 1H, H3a), 2.93 (d, J = 12.0 Hz, 1H, H3b), 2.78 [s, 6H, N(CH_3_)_2_]. ^13^C NMR (75.4 MHz, DMSOd_6_): δ 168.8, 148.9, 134.7, 134.5, 130.8 (JC-F = 8.3 Hz), 126.1, 125.2, 115.2 (JC-F = 21.3 Hz), 111.9, 111.6, 46.7, 40.0, 30.9. HRMS: *m*/*z* (M + H)^+^ calcd. for C_17_H_18_FN_2_OS 317.1120, found, 317.1120.

#### 3.4.12. 5-(4′-Fluorophenyl)-3,5-dihydro-4,1-benzothiazepin-2-(1*H*)-one (**32**)

Following the General procedure 4.4, reaction of **30** (0.36 mmol, 109 mg) [methyl thioglycolate (2.16 mmol, 229 mg), TFA (5.04 mmol, 575 mg, 388 μL)] yielded **32** (12 mg, 12%) as a white solid. Mp: 251–254 °C. ^1^H NMR (300 MHz, DMSOd_6_) δ 7.47 (m, 2H, Ar), 7.46–7.23 (m, 3H, Ar), 7.08 (m, 3H, Ar), 6.98 (m, 1H, Ar), 5.66 (s, 1H, H5), 3.29 (d, J = 12.2 Hz, 1H, H3a), 3.02 (d, J = 12.2 Hz, 1H, H3b). ^13^C NMR (75.4 MHz, CDCl_3_): δ 170.0, 157.6 (d, J = 245 Hz), 136.2, 134.9, 133.2, 131.1 (d, J = 8.0 Hz), 128.9, 128.8, 127.9, 124.3, 115.8 (d, J = 21.0 Hz), 47.2, 31.5. HRMS: *m*/*z* (M + Na)^+^ calcd. for C_15_H_12_FNNaOS 296.0621, found, 296.0621.

#### 3.4.13. 5-(2′-Methoxyphenyl)-3,5-dihydro-4,1-benzothiazepin-2-(1*H*)-one (**33**)

Following the General procedure 4.4, reaction of **31** (0.98 mmol, 308 mg) [methyl thioglycolate (5.88 mmol, 624 mg), TFA (13.72 mmol, 1.56 g, 1.06 mL)] yielded **33** (23 mg, 8%) as a white solid. Mp: 216–218 °C. ^1^H NMR (300 MHz, CDCl_3_) δ 7.76 (d, J = 9.2 Hz, 1H, Ar), 7.63–6.79 (m, 8H, Ar), 6.20 (s, 1H, H5), 3.68 (s, 3H, OCH_3_), 3.33 (d, J = 12.1 Hz, 1H, H3a), 2.99 (d, J = 12.1 Hz, 1H, H3b). ^13^C NMR (75.4 MHz, CDCl_3_): δ 170.3, 156.7, 136.4, 134.9, 130.8, 129.4, 128.4, 128.4, 127.8, 125.4, 124.0, 120.6, 111.1, 55.7, 40.1, 31.6. Anal. C_16_H_15_NO_2_S (C,H,N).

### 3.5. Synthesis of (4-Aminopyridin-3-yl)(phenyl)methanol (**36**)

Similar to what we have described [36]. To a solution of intermediate **35** (0.63 mmol, 179 mg) in 1,4-dioxane (18 mL), KOH_aq_ 2 M (18 mL) was added, and the mixture was refluxed for 24 h. Then, reaction mixture was cooled down to rt and extracted with ethyl acetate (3 × 50 mL). The combined organic layer was dried over anhydrous Na_2_SO_4_, filtered, and evaporated, obtaining **36** as a yellow solid (132 mg, >99%). ^1^H NMR (300 MHz, CDCl_3_) δ 8.00 (d, J = 5.7 Hz, 1H, Ar), 7.86 (s, 1H, Ar), 7.37–7.30 (m, 5H, Ar), 6.42 (d, J = 5.7 Hz, 1H, Ar), 5.78 (s, 1H, CHOH), 4.76 (s, 3H, OH, NH_2_).

### 3.6. Synthesis of Methyl 2-[(4-Aminopyridin-3-yl)(phenyl)methylthio]acetate (**37**)

Similar to what we have described [36]. To a solution of intermediate **36** (445 mg, 2.22 mmol) in dry CH_2_Cl_2_ (25 mL), methyl thioglycolate (13.3 mmol, 1.41 g) and TFA (10 mL) were added under Ar. The mixture was stirred at 45 °C for 22 h. Then, reaction mixture was concentrated, dissolved in CH_2_Cl_2_ (50 mL), and washed subsequently with saturated NaHCO_3_ (50 mL), NaOH (1N, 50 mL), and brine (50 mL). The organic layer was dried over Na_2_SO_4_, filtered, and evaporated. The crude was purified by automatized flash chromatography using ethyl acetate/n-hexane mixtures as eluent to give **37** (144 mg, 25%). ^1^H NMR (300 MHz, CDCl_3_) δ 8.12 (d, J = 5.7 Hz, 1H, Ar), 7.89 (s, 1H, Ar), 7.51 (m, 2H, Ar), 7.41–7.32 (m, 3H, Ar), 6.56 (d, J = 5.7 Hz, 1H, Ar), 5.35 (s, 1H, CH), 5.08 (s, 2 H, NH_2_), 3.74 (s, 3H, OCH_3_), 3.22 (d, J = 16.8 Hz, 1H, H2a), 3.14 (d, J = 16.8 Hz, 1H, H2b).

### 3.7. Synthesis of 5-Phenyl-3,5-dihydropyrido[4,3-e][1,4]thiazepin-2-(1H)-one (**38**)

Similar to what we have described [36]. To a solution of intermediate **37** (150 mg, 0.52 mmol) in dry THF (23 mL), LiHMDS (1.04 mmol, 1.0 M in hexane, 1.04 mL) was added dropwise at −78 °C under Ar and stirred at this temperature for 15 min. Then, reaction mixture was allowed to warm up to rt for 24 h. Reaction was terminated by adding a saturated solution of NH_4_Cl (50 mL) and extracted with CH_2_Cl_2_ (3 × 30 mL). The combined organic layer was washed with brine (3 × 50 mL), dried over anhydrous Na_2_SO_4_, filtered and evaporated, to obtain an orange solid (135 mg, 90%). Mp: 138–143 °C. ^1^H NMR (300 MHz, CDCl_3_) δ 8.52 (d, J = 5.7 Hz, 1H, Ar), 8.30 (s, 1H, Ar), 7.48–7.37 (m, 5H, Ar), 6.93 (d, J = 5.7 Hz, 1H, Ar), 5.63 (s, 1H, H5), 3.30 (d, J = 13.2 Hz, 1H, H3a), 3.15 (d, J = 13.2 Hz, 1H, H3b). ^13^C NMR (75.4 MHz, CDCl_3_): δ 170.4, 151.0, 150.0, 144.0, 136.4, 129.3, 129.0, 128.9, 128.4, 117.4, 46.3, 31.6. HRMS: *m*/*z* (M + H)^+^ calcd. for C_14_H_13_N_2_OS 257.0743, found, 257.0742.

### 3.8. Experimental Use of Animals

Maximum efforts were made to reduce the number of pregnant rats sacrificed for culturing cortical embryonic neurons and their suffering, following the guidelines of the EU Council Directive. Experiments were approved by the Ethics Committee of the Universidad Autónoma de Madrid.

### 3.9. Cell Cultures

The neuroblastoma SH-SY5Y cell line was obtained from the American Type Culture Collection (ATCC-CRL-2266), maintained and cultured in a 1:1 mixture of MEM and F12 media with 10% of fetal bovine serum (FBS), according to procedures previously described [36]. For fluorescence-based [Ca^2+^]_c_ measurements, SH-SY5Y cells were seeded in clear-bottomed, 96-well black plates at a density of 40,000 cells/well. For viability assays, SH-SY5Y cells were seeded in 48-well plates at a density of 70,000 cells/well following the procedure recently reported [44]. Rat cortical embryonic neurons were prepared from 19-day-old embryos directly seeded in the plate well treated with poly-d-lysine in neurobasal medium (Gibco 21103, Madrid, Spain) supplemented with 1% of l-glutamine (Sigma, G6392), 0.5% of penicillin/streptomycin and 0.1% of gentamincine. In fluorescence-based [Ca^2+^]_c_ measurement, neurons were seed at clear-bottomed, 96-well black plates at a density of 30,000 cells/well. For viability assays, the rat cortical neurons were also seeded in 48-well plates at a density of 40,000 cells/well. All the biological assays were conducted under sterile conditions.

### 3.10. Per se Toxicity Assay

The per se toxicity of the new synthetized compounds was assessed in both cell cultures by the method of MTT reduction. After 24 h of seeding, SH-SY5Y cells were preincubated with compounds at different concentrations (1, 3, 10, and 30 μM) for 24 h. Afterwards, MEM/F12 medium was replaced by one fresh with 1% FBS with the compounds at the same concentration. The viability of cultures was measured 20 h later, following a procedure recently described [44]. In the case of rat cortical neurons, 7 days after seeding, compounds were added to the wells to reach a final concentration of 1, 3, 10, and 30 μM. Forty-eight hours later, neuron viability was measure by the same procedure as the one described for SH-SY5Y cells.

### 3.11. Blockade of Cytosolic Ca^2+^ Fluctuations Assay

To assess the blocking effect of compounds on the VGCC, we used SH-SY5Y cells charged with the Ca^2+^-sensitive fluorescent dye Fluo-4/AM at 10 μM in Krebs-HEPES buffer, following a protocol recently described [44]. Compounds were incubated at the desired concentrations (1, 3, 10, and 30 μM). The Ca^2+^ entry was favored by addition of K^+^ 70 mM to the extracellular medium. To evaluate the blocking effect of compounds on NMDA receptors, we have followed a protocol recently reported by our research group [45]. Rat cortical neurons were charged with Fluo-4/AM at 3 μM in Krebs-HEPES buffer and compounds incubated at same concentrations than before. To stimulate Ca^2+^ entry through the NMDA-sensitive glutamate receptors, NMDA was applied at 10 μM. In both cases, fluorescence oscillations as result of the [Ca^2+^]_c_ increases were real-time monitored in a multi-well fluorescence plate reader (FluoStar Optima, BMG, Germany), with excitation and emission wavelengths of 485 and 520 nm, respectively. Data were calculated as fluorescence increase (ΔF), according to the formula:
$$\Delta F=\frac{\left(Fx-Fo\right)\;}{\left(Fmax-Fmin\right)}$$
where *F_x_* is the maximum fluorescence obtained after Ca^2+^ entry stimulation, *F_0_* is the averaged fluorescence before, *F_max_* is the maximum fluorescence of the well obtained by treating cells with 5% Triton X-100, and the *F_min_* is the minimum fluorescence of well obtained by adding MnCl_2_ 1 M to the previously Triton-lysed cells. The Δ*F* data were normalized respect to the control (Δ*F_c_*). Data are presented as percentage of blockade (100% Δ*F*) at the concentration of 10 μM, or as IC_50_, i.e., the concentration of compound capable of halving the Δ*F_c_* (%Δ*F* = 50).

### 3.12. Neuroprotection Assays

The effect of compounds on the SH-SY5Y cells and rat cortical neurons viability compromised of different toxics was also evaluated by method of MTT reduction [46]. Twenty-four hours after seeding, SH-SY5Y cells were preincubated with compounds at the concentration of 10 μM for 24 h. Afterwards, MEM/F12 medium was replaced by one fresh with 1% FBS, compounds at the same concentration, and rotenone/oligomycin A administered to media at the concentration of 30/10 μM, respectively. The cultures’ viability was measured 20 h later, following the MTT procedure recently described [44]. As reference neuroprotective compound, melatonin 10 nM was selected. In the case of the rat cortical neurons, 7 days after seeding in 48-well plates, neurons were preincubated adding the compounds at a final concentration of 10 μM for 24 h. Then, toxic stimuli of 50 µM of glutamate was added to the wells. After 20 h, cell viability was measured by the MTT procedure [44]. Here, the reference protective compound used was memantine (30 nM).

### 3.13. Blood–Brain Barrier Permeation Assay (PAMPA)

The parallel artificial membrane permeation assay (PAMPA) was used to predict the capacity of compounds to cross the blood–brain barrier (BBB) by passive diffusion, in a similar manner as previously described [43]. Compounds were dissolved in 10 mM phosphate-buffered saline (PBS) buffer (pH 7.4) to the concentration of 100 µM. The filter membrane of the donor 96-well plate (Multiscreen IP sterile clear plate PDVF membrane, pore size 0.45 μM) was impregnated with 4 µL of porcine brain lipid (PBL) (Avanti Polar Lipids, Inc., Alabaster, AL, USA) in dodecane (20 mg/mL), and after 5 min, 180 µL of each compound solution was added. The acceptor 96-well plate (Multiscreen, Millipore Corp. Burlington, MA, USA) was filled with 180 µL of PBS and 1% of DMSO. Finally, the donor filter plate was carefully put on the acceptor plate to form a “sandwich”, which was left undisturbed for 4 h at rt. After incubation, an UV plate reader determined the concentration of compounds in both acceptor and donor wells (150 µL/well) as the maximum absorption wavelength of each compound. Concentration of the compounds in the donor and acceptor well, and equilibrium concentration were calculated from the standard curve and expressed as the permeability (Pe), according to the following formula:
$$\log\mathrm{Pe}=\log\left\{C\times -\ln\left(1-\frac{\left[\mathrm{drug}\right]\mathrm{acceptor}}{\left[\mathrm{drug}\right]\mathrm{equilibrium}}\right)\right\}$$
$$\mathrm{where}C=\frac{\mathrm{Vdonor}\times \mathrm{Vaceptor}}{\left(\mathrm{Vdonor}+\mathrm{Vaceptor}\right)\times \mathrm{Area}\times \mathrm{Time})}$$

Results are expressed as mean ± SEM of three different experiments in duplicate.

### 3.14. Data Analysis

To calculate the necessary sample size (*n*) for each experiment, taking a confidence interval (*Z*) of 95% and an error margin admitted (*e*) of 5%, we applied the formula *n = Z^2^·s^2^/e^2^*, where the averaged standard deviation (*s*) in our similar previous experiments using neuroblastoma cells was 0.04. Thus, a sample size of a minimum of three experiments was necessary. Comparisons between control and test groups were executed by a one-way analysis of variance (ANOVA), followed by Dunnett or Newman–Keuls post hoc test, using the GraphPad Prism 5.0 for Mac OSX software, and considered statistical different when *p* < 0.05.

## 4. Conclusions

A series of new **CGP37157** derivatives with improved physicochemical properties were synthesized, and their pharmacological properties were evaluated. Their ability to buffer the neuronal Ca^2+^ by controlling the mitochondrial regulatory machinery, and thus protecting against neurotoxic stimuli derived from Ca^2+^ overload, make them promising potential neuroprotective drugs. Despite their modest capacity to mitigate neuronal Ca^2+^ elevation, the first subgroup of new 4,1-benzothiazepines present a significant protection against oxidative stress, and a slight capacity to mitigate neuronal Ca^2+^ elevation. Among these derivatives, compound **18** stands out thanks to its ability to reduce Ca^2+^ overload induced by both K^+^-evoked depolarization and NMDA stimulation, as well as its neuroprotective profile against both R/O-induced oxidative stress and high glutamate-exerted excitotoxicity. Notably, compound **18** is the closest analogue to **CGP37157**, with an added aromatic nitrogen in the *para* position of the pending phenyl ring. This slight modification allowed the compound to overcome the solubility problems of **CGP37157**, keeping its pharmacological properties similar. Regarding to the family of 7-dimethilaminobenzothiazepines, the derivative **21**, which possesses an o-isopropyl substituent at the pending phenyl ring, showed the most promising regulatory capacity of [Ca^2+^]_c_ levels, correlating with an interesting reduction of the neuronal death in an excitotoxicity scenario. Again, compound **21** is the closest analogue to our previous head compound **ITH12575** [35], which is currently under investigation for several pathologies affecting mitochondrial function, such as acute traumatic cerebral damage. In the case of the new derivatives lacking substitution at C7, compounds **32** and **33**, although they were not good at regulating Ca^2+^ overload, they have shown a significant neuroprotection against glutamate. Finally, the 4-aminopyridine-fused thiazepine **38**, has a very interesting protective profile against the neurotoxic stimuli used, even without presenting good regulatory capability of neuronal Ca^2+^. Almost all the new compounds feature a highly probable passive diffusion through the BBB, which is a crucial fact for any drug targeting the CNS. In fact, the most promising compounds, **18** and **21**, have a Pe value higher than the threshold value of 40 nm/s (Figure 6). Excluding compounds **13** and **14**, which surprisingly were slightly soluble and toxic, the rest of the new compounds presenting diverse substitution have shown interesting neuroprotective profiles. Overall, we highlight compounds **18** and **21**, as promising optimized benzothiazepine derivatives for further studies to define their neuroprotective properties.

## Data Availability

Data supporting obtained results can be obtained from the authors upon request.

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
