# Peer review of "Synthesis and Biological Assessment of 4,1-Benzothiazepines with Neuroprotective Activity on the Ca2+ Overload for the Treatment of Neurodegenerative Diseases and Stroke"

_molecules, 2021, doi:10.3390/molecules26154473_

Round 1

Reviewer 1 Report

The draft manuscript entitled "Synthesis and Biological Evaluation of 4,1-Benzothiazepines with Neuroprotective Activity on Ca2 + Overload for the Treatment of Neurodegenerative Diseases and Stroke" is quite interesting and attractive to readers. This is very important work in terms of organic synthesis and biological evaluation. The paper is acceptable for publication by following the recommendations below:

  1. 1. All new compounds must be characterized by HRMS or by microanalysis
  2. 2. Page4- line 123 : change « p-chloroanyline » by « p-chloaniline »
  3. 3. In 1H NMR of compound 5 and 6. Change « 2.94 [p, J = 6.9 Hz, 1H, CH(CH3)2] » by « 2.94 [m, J = 6.9 Hz, 1H, CH(CH3)2] »
  4. 4. Why in the « supporting information » part, we don’t find the spectral data of the compounds obtained.

Author Response

-Reviewer 1

The draft manuscript entitled "Synthesis and Biological Evaluation of 4,1-Benzothiazepines with Neuroprotective Activity on Ca2 + Overload for the Treatment of Neurodegenerative Diseases and Stroke" is quite interesting and attractive to readers. This is very important work in terms of organic synthesis and biological evaluation.

Thank you very much

The paper is acceptable for publication by following the recommendations below:

  1. All new compounds must be characterized by HRMS or by microanalysis

Done. All the final compounds are now characterized by HRMS (13, 16, 17, 18, 21, 26, 28, 32, 38) or microanalysis (12, 14, 15, 26, 27, 33)

  1. 2. Page4- line 123 : change « p-chloroanyline » by « p-chloaniline »

Corrected

  1. 3. In 1H NMR of compound 5 and 6. Change « 2.94 [p, J = 6.9 Hz, 1H, CH(CH3)2] » by « 2.94 [m, J = 6.9 Hz, 1H, CH(CH3)2] »

Corrected. Similarly “[p,…” have been replaced by “[m,...” in 1H NMR of compounds 20, 12, 13 and 21

  1. 4. Why in the « supporting information » part, we don’t find the spectral data of the compounds obtained.

Spectral data (1H-NMR and 13C-NMR) have now been included in the Supporting Information.

Thank you very much for all of your comments and suggestions.

Reviewer 2 Report

In this manuscript, by Viejo et al., a series of new benzothiazepines were synthesized, and their pharmacological properties were evaluated in fluorescence, MTT and BBB permeation assays. The manuscript flows well; however I have some concerns about certain aspects of the research.

Major concerns:

  1. “2.2.2. Effect of compounds on the neuronal Ca2+ overload”; this title does not fit with the content. The compounds showed effect on membrane depolarization-induced Ca2+ influx in SH-SY5Y cells. However, it is a completely different concept from Ca2+ overload. The results only indicated inhibitory activities of several compounds on HVA calcium channels expressed in SH-SY5Y cells.
  2. Again in 2.2.2., as is known, both N-type and L-type VGCC are endogenously expressed in SH-SY5Y cells. Generally, in fluorescence/electrophysiological assays, nifedipine has been used to block L-type currents, so that effects on N-type currents could be measured; and ω-conotoxins like CVID/MVIIA has been used to block N-type currents for measuring of effects on L-type responses. However, it is not applied in this study. Here, my question is, when the IC50s were measured for these compounds, do they give partial or full inhibition of the SH-SY5Y Ca2+ current? Also, it does not make much sense to me to use nifedipine as the control as it surely doesn’t fully block SH-SY5Y Ca2+currents. 

Minor comments:

  1. Line 39, “Between the second ones”, should be “Among the second ones”;
  2. Line 63, it is not accurate to call NMDAR and AMPAR “Ca2+ ionic channels”, could be changed to “Ca2+ permeable channels”;
  3. In figure 5, it would be better to present the results as percentage of protection instead.

Author Response

In this manuscript, by Viejo et al., a series of new benzothiazepines were synthesized, and their pharmacological properties were evaluated in fluorescence, MTT and BBB permeation assays. The manuscript flows well;

Thank you very much

however I have some concerns about certain aspects of the research.

Major concerns:

  1. “2.2.2. Effect of compounds on the neuronal Ca2+ overload”; this title does not fit with the content. The compounds showed effect on membrane depolarization-induced Ca2+ influx in SH-SY5Y cells. However, it is a completely different concept from Ca2+ overload. The results only indicated inhibitory activities of several compounds on HVA calcium channels expressed in SH-SY5Y cells.

Thank you very much. We agree with the reviewer’s suggestion and thus we have changed the subchapter title. In the paragraph, we have checked that an explanation on the relationship between depolarization or NMDAr stimulation with calcium overload and neurodegeneration progression had been included.

  1. Again in 2.2.2., as is known, both N-type and L-type VGCC are endogenously expressed in SH-SY5Y cells. Generally, in fluorescence/electrophysiological assays, nifedipine has been used to block L-type currents, so that effects on N-type currents could be measured; and ω-conotoxins like CVID/MVIIA has been used to block N-type currents for measuring of effects on L-type responses. However, it is not applied in this study. Here, my question is, when the IC50s were measured for these compounds, do they give partial or full inhibition of the SH-SY5Y Ca2+ current?

Thank you very much. As mentioned in the experimental part (3.11), compounds were assayed at 1, 3, 10 and 30 microM to describe their IC50, defined as the concentration of compounds capable of halving the calcium entry stimulated by 70K or NMDA. Thus, maximal concentration tested in all the compounds was 30 uM. At this concentration, compounds did not fully block calcium entry. Higher concentrations were not probed due to solubility issues.

  1. Also, it does not make much sense to me to use nifedipine as the control as it surely doesn’t fully block SH-SY5Y Ca2+currents.

Calcium measurements were monitored by fluorescent experiments. Nifedipine was used as a standard of the Fluo-4-probed fluorescent assays in SH-SY5Y cells stimulated with 70 mM K, not because we seeked a full VGCC blockade, but only as a reporter of the correct development of the experiment, and the viability of the cell line batch.

Minor comments:

  1. Line 39, “Between the second ones”, should be “Among the second ones”

Corrected

  1. Line 63, it is not accurate to call NMDAR and AMPAR “Ca2+ ionic channels”, could be changed to “Ca2+ permeable channels”

Corrected

  1. In figure 5, it would be better to present the results as percentage of protection instead

Figure 5 depicted %Cell viability to show the loss of cell viability elicited by the toxic stimuli (black bars), either rotenone plus oligomycin A (R/O) or glutamate. In these graphs, the ability of some compounds to counteract such loss of viability in a statistically significant fashion is clearly presented. We now include a table with the percentage of protection of the compounds presenting statitistically significant differences with the control.

Thank you very much for all of your suggestions and comments.

Reviewer 3 Report

The present research article entitled “Synthesis and biological assessment of 4,1-benzothiazepines with neuroprotective activity on the Ca2+ overload for the treatment of neurodegenerative diseases and stroke” by Viejo et al. demonstrates medicinal chemistry approach to validate in vitro neuroprotective efficacy of novel synthesized benzothiazepine derivatives with specific emphasis on Ca2+ ion flux via NMDA and AMPA receptors with implications towards neurodegenerative diseases. Authors used standard chemistry synthesis, toxicological and pharmacological methods to characterize their compounds as neuroprotective agents. There are some queries/comments that need to be addressed by the authors, as follows:

#1. Although, authors mentioned that they characterized novel benzothiazepines analogues as NCLX blockers, however, authors need to demonstrate the specificity and selectivity over other mitochondrial Ca2+ exchangers such as H+/Ca2+ exchanger. Authors can use molecular docking or valid drug designing computational approaches to calculate the binding affinity of their synthesized molecules.

#2. Authors need to estimate the mitochondrial ATP levels, because novel molecule treatment could alter the cellular energy homeostasis due to mitochondrial oxidative stress.

#3. Authors used MTT assay to assess cytotoxicity, however neuronal apoptosis can be accurately evaluated by using LDH assay, therefore authors need to conduct LDH assay as more reliable and valid toxicity assay, especially in case of primary cortical neurons to demonstrate the neuroprotective efficacy of their molecules.

Author Response

The present research article entitled “Synthesis and biological assessment of 4,1-benzothiazepines with neuroprotective activity on the Ca2+ overload for the treatment of neurodegenerative diseases and stroke” by Viejo et al. demonstrates medicinal chemistry approach to validate in vitro neuroprotective efficacy of novel synthesized benzothiazepine derivatives with specific emphasis on Ca2+ ion flux via NMDA and AMPA receptors with implications towards neurodegenerative diseases. Authors used standard chemistry synthesis, toxicological and pharmacological methods to characterize their compounds as neuroprotective agents.

Thank you very much

There are some queries/comments that need to be addressed by the authors, as follows:

#1. Although, authors mentioned that they characterized novel benzothiazepines analogues as NCLX blockers, however, authors need to demonstrate the specificity and selectivity over other mitochondrial Ca2+ exchangers such as H+/Ca2+ exchanger.

Thank you very much. Although the contribution of H/Ca exchanger is much minor compared with that of NCLX to the calcium fluxes between mitochondria and cytosol in excitable cells, we find interesting the experiments proposed by the Reviewer. We think that they could be performed in the future with optimized analogues selected to further investigate their pharmacological properties.

Authors can use molecular docking or valid drug designing computational approaches to calculate the binding affinity of their synthesized molecules.

Thank you very much. Unfortunately, NCLX protein has not been deposited in the Protein Data Bank yet. Previously, with the goal of aiding our studies with computational techniques, we attempted to build a homology model from the exchanger NCKX6_HUMAN, from UNIPROT, generating the 3D homologous in PHYRE2, but only 56% of the amino acids were modeled with a 90%  accuracy. The ionic channel was modeled from the sequence Q57556 (Y091_METJA), a NCX from Methanocaldococcus jamnaschii, deposited in the PDB, then optimized in Molegro Virtual Docker. We executed a virtual screening of 59283 molecules, but the most efficient ligand of this 3D model resulted to be inactive in the pharmacological experiments used. Thus, we conclude that our homology model was not still valid to predict and confirm the pharmacological activity of our compounds.

#2. Authors need to estimate the mitochondrial ATP levels, because novel molecule treatment could alter the cellular energy homeostasis due to mitochondrial oxidative stress.

Thank you very much. Although these experiments would enrich the manuscript, any alteration of the energy homeostasis due to dramatic changes in ATP levels would produce clear toxic effects and an important loss of cell viability. Here, we demonstrated that, at the selected testing concentrations and especially with the most promising compounds, toxic effects are not observed (Figure 2). Moreover, we prove in Figure 5 that 6 out 12 compounds protected against R/O, a toxic stimulus that indeed provokes ATP synthesis disruption.

#3. Authors used MTT assay to assess cytotoxicity, however neuronal apoptosis can be accurately evaluated by using LDH assay, therefore authors need to conduct LDH assay as more reliable and valid toxicity assay, especially in case of primary cortical neurons to demonstrate the neuroprotective efficacy of their molecules.

MTT assay is a validated method widely used in SH-SY5Y cells and cortical neurons, as shown in these references:

- Hasegawa, K. et al “Promotion of mitochondrial biogenesis by necdin protects neurons against mitochondrial insults” Nat Commun 7, 10943 (2016)

- Ke Wang et al “Harpagide from Scrophularia protects rat cortical neurons from oxygen-glucose deprivation and reoxygenation-induced injury by decreasing endoplasmic reticulum stress”, Journal of Ethnopharmacology, 253, 112614 (2020)

- Trapani, A. et al “Cyto/Biocompatibility of Dopamine Combined with the Antioxidant Grape Seed-Derived Polyphenol Compounds in Solid Lipid Nanoparticles” Molecules 26, 916 (2021)

- Wang, T. et al “Hydroxysafflor Yellow A Attenuates Lipopolysaccharide-Induced Neurotoxicity and Neuroinflammation in Primary Mesencephalic Cultures” Molecules, 23, 1210 (2018)

In our research group, we have used LDH method in the past as a routine method to evaluate neuroprotective drugs, see for instance:

  • Camilo Orozco, Cristóbal de los Ríos et al Pharmacol. Exp. Ther. 2004, 310, 987
  • José Marco-Contelles et al. Med. Chem. 2006, 14, 8176.
  • José Marco-Contelles et al. Med. Chem. 2006, 49, 7607.
  • Rafael León and Cristóbal de los Ríos et al. Med. Chem. 2008, 16, 7759.
  • - Rafael Léon and Cristóbal de los Ríos et al. J. Med. Chem. 2008, 43, 668
  • José Marco-Contelles et al. Med. Chem. 2009, 52, 2724.

However, during those research works, we appreciated some issues related to the experimental protocol necessary to measure LDH, when testing compounds at high concentrations. According to the manufacturer, LDH kit was added to the media and then absorbance of cultures is monitored at 490 nm. Thus, higher concentrations of many tested compounds increased absorbance at that wavelength, because of their own UV-Vis absorption spectra. Thus, this artefact showed that the more compound concentration, the more cell death experimented by the cell culture.

By contrast, this kind of artefact is avoided in the MTT assay, since media is harvested before measurements, then the precipitated and reduced form of MTT is reconstituted in DMSO and monitored at 545 nm. For this reason, since 2010, we have been progressively replacing LDH by MTT to test neuroprotective properties of the newly-synthesized compounds of our research group, see for instance:

  • Cristóbal de los Ríos et al Med. Chem. 2010, 53, 5129.
  • Abdelouhid Samadi et al Med. Chem. 2010, 18, 5861
  • Laura González Lafuente et al ACS Chem. Neurosci. 2012, 3, 519
  • Francisco J. Martínez-Sanz et al ACS Chem. Neurosci. 2015, 6, 1626.
  • Francisco J. Martínez-Sanz et al J. Med. Chem. 2016, 109, 114.

As we evidenced that MTT was a colorimetric method that could fully replace LDH method to assay potential new neuroprotective compounds, we finally decided to use only MTT method during the last years, see for instance:

  • Silvia Lorrio et al ACS Chem. Neurosci. 2013, 4, 1267.
  • Alejandro Romero et al ACS Chem. Neurosci. 2014, 5, 770.
  • Rocío Lajarín-Cuesta et al Med. Chem. 2016, 59, 6265.
  • Dorleta González et al Med. Chem. 2018, 26, 2551
  • Rocío Lajarín-Cuesta J. Med. Chem. 2018, 157, 294.
  • Alejandra Bisi et al. J. Med. Chem. 2019, 163, 394
  • Eva Ramos et al Res. Toxicol. 2021, 34, 1245.

Thank you very much for all of your comments

Round 2

Reviewer 1 Report

The manuscript entitled "Synthesis and Biological Evaluation of 4,1-Benzothiazepines with Neuroprotective Activity on Ca2 + Overload for the Treatment of Neurodegenerative Diseases and Stroke" is quite interesting and attractive to readers. The paper is acceptable for publication.

Reviewer 3 Report

Revised manuscript is provided with acceptable rationale for not including suggested additional experiments, such as ATP measurement and LDH assay instead of MTT. Also, in future, when NCLX protein structure is available, it would be much useful to validate the designing of the compounds using computational tools to define selectivity of the novel compounds.